# Zebrafish Models to Study the Crosstalk between Inflammation and NADPH Oxidase-Derived Oxidative Stress in Melanoma

**DOI:** 10.3390/antiox11071277

**Published:** 2022-06-28

**Authors:** Irene Pardo-Sánchez, Diana García-Moreno, Victoriano Mulero

**Affiliations:** 1Departamento de Biología Celular e Histología, Facultad de Biología, Universidad de Murcia, 30100 Murcia, Spain; irene.pardo1@um.es; 2Instituto Murciano de Investigación Biosanitaria (IMIB)-Arrixaca, 30120 Murcia, Spain; 3Centro de Investigación Biomédica en Red de Enfermedades Raras (CIBERER), Instituto de Salud Carlos III, 28029 Madrid, Spain

**Keywords:** melanoma, oxidative stress, NADPH oxidases, DUOX1, inflammation, macrophages, neutrophils, zebrafish

## Abstract

Melanoma is the deadliest form of skin cancer, and its incidence continues to increase. In the early stages of melanoma, when the malignant cells have not spread to lymph nodes, they can be removed by simple surgery and there is usually low recurrence. Melanoma has a high mortality rate due to its ability to metastasize; once melanoma has spread, it becomes a major health complication. For these reasons, it is important to study how healthy melanocytes transform into melanoma cells, how they interact with the immune system, which mechanisms they use to escape immunosurveillance, and, finally, how they spread and colonize other tissues, metastasizing. Inflammation and oxidative stress play important roles in the development of several types of cancer, including melanoma, but it is not yet clear under which conditions they are beneficial or detrimental. Models capable of studying the relevance of inflammation and oxidative stress in the early steps of melanocyte transformation are urgently needed, as they are expected to help recognize premetastatic lesions in patients by improving both early detection and the development of new therapies.

## 1. Introduction

Melanoma is the fifth most common cancer and the most serious skin cancer. It is estimated that in 2021 there will be more than 100,000 new cases of melanoma in the United States and around 7000 deaths from this disease [1]. The development of melanoma is most common in adults with a median age at diagnosis of 65, though it also affects young people (4.8% from 20 to 34 years old).

Melanoma is caused by the malignant transformation of melanocytes, a type of skin cell that produces melanin to protect against exposure to UV light [2]. In the 1960s, five levels of anatomical invasion of melanoma in the skin were described. The first step of melanoma begins as a proliferation of normal melanocytes to form a nevus, and in this level the melanocytes are still confined to the epidermis. Then, the nevus acquires an atypical growth, and a dysplastic nevus develops. Then, unlimited radial growth begins, followed by a vertical growth phase, which crosses the basement membrane forming a tumor, and, finally, successfully spreads throughout the body as metastatic tumors [3,4]. Most melanoma lesions that only affect the epidermis, the most external layer of skin, are resolved with initial surgery, and the 5-year survival rate, over 93%, is very encouraging. However, when a melanoma involves multiple layers of skin, it can spread to other parts of the body and metastasize [2]. If the melanoma has spread to the lymphatic nodes, the 5-year survival rate decreases, and if the melanoma has spread through the bloodstream to distant parts of the body, the survival rate is even lower, around 25% [5].

Currently, melanoma develops due to multifactorial conditions. It is a complex multistep process, and its understanding is crucial [6]. This process involves not only the transformation of a cell, but also local tumor invasion, vascularization, dissemination, tumor establishment in another site, and growth of the secondary tumor [7]. Melanoma cells should be able to adapt to different microenvironments to invade and metastasize. Crosstalk between multiple pathways points to the importance of understanding the precise role of multiple factors in the tumor microenvironment [8].

It has been proposed that cutaneous melanoma can be divided into four subtypes according to mutation in the three most prevalent mutated genes: BRAF, NRAS, NF1, and Triple WT (wild type), the latter without any of them [9]. Cutaneous melanomas often harbor BRAF mutations (≈50%) and to a lesser degree NRAS mutations (28%). BRAF is the main oncogene found in both malignant melanoma and in benign nevi. BRAF^V600E^ mutations are present in most nevus, so other lesions would have to occur to develop melanoma. However, this fact points to the important role of this oncogene in melanocyte transformation at early stages.

In melanoma, the RAS-RAF-MEK-ERK (mitogen-activated protein kinase, MAPK) and the PTEN-PI3K-AKT (AKT) signaling pathways are constitutively activated through multiple mechanisms, and play several key roles in the development and progression of melanoma [10]. Activation of MAPK pathway culminates in the regulation of gene transcription in the nucleus by the extracellular signal-regulated kinase ERK, which phosphorylates several cellular substrates, enabling proliferation [11]. NRAS and BRAF molecules belong to the MAPK signal transduction pathway, which play a key role in regulating cell growth, survival, and cell proliferation. In melanocytes, BRAF induces the activation of MEK kinase, which activates ERK, the end effector of the MAPK cascade, via phosphorylation. This results in continuous stimulation of cell proliferation and tumor growth [12].

The second major pathway of cell growth regulation is the signal transduction AKT cascade depending on RAS [13]. It has been demonstrated that the activation of AKT1 results in the development of more metastatic melanomas in mice [14]. In addition, microphthalmia-associated transcription factor (MITF) is involved in the control of proliferation and differentiation of melanocytes, and is also associated with melanoma development and progression [15]. In melanoma, constitutive activation of ERK is associated with a marked degradation of MITF [12]. The role of MITF is complex. Melanoma cells expressing MITF at a high level can either differentiate or proliferate; however, low activity of MITF is related to stem cell-like or invasive potential [16].

To study the contribution of these signaling pathways to melanoma progression, human samples and different animal models have been used. In recent years, the zebrafish has been recognized as an animal model for the study of several diseases, including cancer and notably melanoma [17].

## 2. Zebrafish as a Research Model

The zebrafish is an established model organism for studying developmental biology and molecular genetics. Currently it is a powerful model organism in biomedical research, and its popularity has increased in recent years mainly due to its advantages over other models [18] (Figure 1). Its small size and ease of maintenance coupled, with its high fecundity that allows many replicates (around 200 eggs/couple/week), and its rapid development, with organs fully formed 48 h post-fertilization (hpf), are the main advantages offered by zebrafish. Moreover, zebrafish share 71% of genetic similarity with humans, and 82% of human disease-related genes can be linked to at least one zebrafish orthologue [19]. In melanoma studies, the translucency of zebrafish larvae is a key advantage that allows in vivo imaging of the process at the very early stages of oncogenic transformation and studying their interaction with immune cells. Finally, another main advantage is its amenability for in vivo chemical screening.

Most the analyses of the metastatic process have been performed using in vitro cell cultures [20,21] or in mice models [22,23]. Cell cultures neglect the complexity of the process, and it is not possible to observe the dissemination of these cells or the interaction with the tumor microenvironment. To address these issues, in vivo models are mandatory. But in mice, it is difficult to study the first steps of tumor development; small lesions are impossible to observe in vivo due to the depth of the tissues that mice have and usually have to be sacrificed. Therefore, the metastatic process is usually assessed at the endpoint and not during the development (Figure 1). Furthermore, the number of individuals used in each experiment is limited, and the statistical power cannot be as high as it should be [24]. Recently, the zebrafish has emerged as a complementary model to overcome these disadvantages [25,26], offering alternative options to study the processes involved in the development of melanoma (Figure 2).

### 2.1. Genetics Models

To address the study of melanoma, zebrafish models expressing different human oncogenes were developed. A model expressing transgenic human BRAF^V600E^ combined with the tumor protein P53 (TP53) mutation was first established [27]. Later, to study the role of NRAS using zebrafish, transgenic human NRAS^Q61K^ mutants were also described [28]. More recently, the relevance of the zebrafish model in melanoma research was highlighted by the identification of the loss of SPRED1 in mucosal melanoma [29] and the role of anatomical position in determining oncogenic specificity [30].

### 2.2. Xenograft in Larvae

Microinjection of human and mouse melanoma cells into zebrafish embryos is a widely used technique to study migration and invasion processes. It is usually performed at 48 hpf, when the adaptive immune response has not yet been established [31,32], so this method does not require immunosuppression. First, the cells must be labelled with a fluorescent probe, such as cM-Dil, to allow the monitoring of cancer cells [33]. After injection of the cells into the yolk sac, the ability of the cancer cells to proliferate or invade different tissues through the larvae can be followed and measured [34]. Considering the possibility of using transgenic zebrafish lines with fluorescently labelled immune cells, this model allows the study of essential interactions between tumor cells and the host immune microenvironment in vivo [4].

### 2.3. Allograft in Adults

The study of melanoma in adult zebrafish allows in vivo analysis of tumor engraftment and migration after transplantation. This technique consists of injecting disaggregated melanoma cells from a donor fish into the dorsal subcutaneous cavity of a recipient fish to visualize their proliferation and dissemination in vivo [35]. One of the pitfalls of this technique is the immune rejection of transplanted cells in adults, which makes the immunosuppression of zebrafish by irradiation necessary [36]. Another drawback for transplanting cells into adults is that the natural pigmentation of the fish does not allow most of the cellular processes to be followed. To solve this, unpigmented zebrafish models, named “casper”, were developed to study cancer mechanisms such as angiogenesis, migration, invasion, or tumor growth in adult zebrafish. The casper zebrafish model is a combination of two mutations: the first one, *nacre,* which has a mutation in *mitfa*, which regulates neural crest derived pigment [37], and results in a loss of melanoblast and mature melanocytes; while the second mutation is *roy,* that affects the *mpv17* gene, and results in a severe disruption of melanocyte numbers and patterning and loss of iridophores [38]. Using melanin or GFP as markers, melanoma cells are easier to visualize and track. Furthermore, combining this model with transgenic lines that label immune cells, such as neutrophils [39] or macrophages [40,41] or vasculature [42], helps to gain a more detailed understanding of how the transformed cells interact with the immune system and their niche. It is possible to track not only innate immune cells, but also some adaptative immune cells, such as T cells with the lck *promoter* [36], or specific CD4^+^ T cells [43]. Unfortunately, there are no available lines with labelled CD8^+^ T lymphocytes and NK cells to track them in vivo and study their relevance in tumor immunosurveillance.

One of the most important applications of this model is the possibility to test drugs in vivo and in an easier and cheaper system than mouse models. For example, it has been used to test long term administration of drugs in adult casper zebrafish intraperitoneally transplanted with a zebrafish melanoma cell line (ZMEL1), and a tumor reduction could be observed after the oral administration of the drug [44].

### 2.4. Xenograft in Adults

Transplantation of human tumor cells into zebrafish is easy in larval stages where the immune system is not fully developed, but in adults it requires laborious work and transient methods of immune suppression that limit engraftment and survival of the tumor, or it does not reproduce the characteristics of these malignancies [45]. Xenograft models in adult zebrafish have previously been used to study pancreatic cancer progression [46] and tumor cell intravasation in T-lymphoblastic lymphoma [47]. More recently, immunodeficient zebrafish were generated to visualize human cancer and therapy responses [48]. This fish model has mutations in *prkdc* and *il1rga* genes that result in a lack of adaptive immunity and NK cells. Notably, this model is grown at 37 °C, allowing engraftment of a wide array of human cancers injected in the peritoneal cavity, including patient-derived xenografts, and the study of their dynamics and therapy responses.

### 2.5. Early Transformation

A powerful method has been developed to follow the melanoma initiation cells at very early stages. This is possible due to the knowledge about the expression of *crestin* in the neural crest during embryogenesis; it is no longer expressed after 72 hpf and is only re-expressed in melanoma tumors when melanoma precursor cells re-initiate an embryonic neural crest signature [49]. Therefore, the *crestin*–GFP model allows the following of melanoma development when a tumor starts at very early stages in vivo [50].

### 2.6. The MiniCoopR System

A useful tool for melanoma research has recently been developed to study the effect of a gene on tumor initiation. The MiniCoopR method consists of a plasmid that allows the gene of interest (GOI) to be place under the *mitfa* promoter and also contains a *mitfa* minigene, which combined with the use of casper zebrafish, allows one to easily follow which cells transform because they recover melanin expression [51]. For example, this method has identified the histone methyltransferase SETDB1 as capable of accelerating melanoma formation [52].

The second most common oncogene driving malignant melanoma is NRAS. Recently, a zebrafish model has been generated to study NRAS mutant melanoma using the MiniCoopR vector (*mcr:NRAS*) [53]. In this model, tumor development occurs without mutations in TP53, and the development of melanoma is faster than in previous lines, such as the BRAF and NRAS stable transgenic lines [27,28].

In summary, the unique advantages of zebrafish for in vivo imaging together with the ease of genetic manipulation and the high throughput drug screening make this model a key tool for study of cancer, more specifically melanoma [54]. Zebrafish facilitate the study of melanoma initiation, using casper zebrafish or the recent zebrafish model *crestin:eGFP*, including the monitoring of the development, tumor growth, and metastasis in adult stages. It also makes it easier than other models to study the invasiveness potential and drug resistance of patient tumor cells using xenografts in larvae (Avatar models), and it makes easier the generation of genetic mutants to study the impact of a GOI on melanoma development. In addition, the recent development of the MinicoopR system makes it possible to study the impact of a GOI on tumor biology from early initiation to tumor development.

## 3. Inflammation in Melanoma

Melanoma is one of the most immunogenic types of cancer [55]. The immunogenicity of a tumor is the capacity to induce adaptive immune responses that can prevent its growth [56]. In addition, many studies support the concept that innate immunity plays a crucial role in the development, growth, and prognosis of cutaneous malignant melanoma [57]. A major field of study in melanoma research is the behavior and impact of innate and adaptive immune cells in the development of melanoma.

Inflammation is the response to cellular damage by infectious agents, toxins, or physical stress, such as radiation or previous injuries. The crosstalk between inflammation and cancer has been recognized since 1909, when Ehrlich proposed the “magic bullet” theory, which is now considered a precursor to chemotherapy [58]. Inflammatory cells and signals contribute to cancer mechanisms from the initiation stage through tumor promotion and progression until the development of cancer [59]. Chronic inflammation may cause genomic instability and DNA damage resulting in oncogenic activation or inactivation of tumor suppressors, promoting cancer. The tumor microenvironment has been shown to be crucial in the development of malignancy. In addition, cancer cells can promote an inflammatory microenvironment that supports tumor cell proliferation [60]. Inflammation can also drive other processes that help tumor growth, such as angiogenesis [61].

### 3.1. Role of Macrophages

Macrophages are crucial players in melanoma growth and survival. Initially assumed to be involved in anti-tumor immunity, they have been shown to promote cancer initiation, stimulate angiogenesis, and suppress antitumor immunity during malignant progression [62]. Important evidence for this is the strong association between increased macrophage density and poor survival in glioblastoma, hepatocellular, thyroid, or lung cancers [63,64,65,66].

In general, macrophages can be classified as M1 or M2 activation phenotypes, depending on the molecules that activate them and their different metabolic programs, which can influence different inflammatory responses [67]. In the context of cancer, M1 macrophages predominantly play a role in antitumor immunity, while M2 tumor-associated macrophages (TAM) play a role in immunosuppression and tumor immune escape.

The ability of melanoma exosomes to directly polarize macrophages, which would be expected to promote different pro-tumor functions, has recently been investigated. According to the polarization factors identified, such functions may include: stimulating TAM polarization; tumor growth and metastasis; recruiting other immunosuppressive cell types, such as T regulatory cells (Tregs) and tumor-associated neutrophils (TANs); angiogenesis; and promoting immune suppression [68]. Macrophages have also been shown to directly associate with growing tumor vessels and enhance tumor vascularization [69].

Using zebrafish allograft models, it has been shown that chronic inflammation induced by Spint1a deficiency facilitates oncogenic transformation. These results may be of clinical relevance, as SPINT1 correlates with markers of aggressiveness, poor prognosis, and tumor macrophage infiltration in human cutaneous melanoma [70].

The presence of TAMs correlates with poor prognosis in a wide range of cancers. However, most commonly used models have serious limitations, usually interfering data from fixed samples or at the endpoint of the experiment [71]. To help with this limitation, zebrafish emerged as an incredible model that showed valuable optical properties to facilitate in vivo imaging and helping track the interaction between the immune system and cancer cells. For example, interactions between TAMs and tumor cells have been observed thanks to these unique imaging properties that zebrafish offer in vivo. Thus, it was shown that macrophages are recruited to the tumor site and are able to transfer cytoplasm to melanoma cells, thereby increasing tumor cell motility and promoting tumor cell dissemination [71].

### 3.2. Role of Neutrophils

Neutrophils are also a crucial component of the inflammatory response to tumors. New evidence indicates that tumors may manipulate TANs to create different phenotypic and functional polarization states able to modify tumor behavior [72]. Depending on specific factors derived from the tumor, the polarization of neutrophils leads to different phenotypes. Transforming growth factor-β (TGF-β) and granulocyte colony-stimulating factor (G-CSF) polarize TANs towards a pro-tumorigenic phenotype and promote metastasis formation by regulating transcription factors, such as the inhibitor of DNA binding 1 or interferon regulatory factor 8, which control the immunosuppressive functions of TANs [73,74,75,76].

Moreover, it has been suggested that neutrophils have the ability to influence CD8^+^ T cells in infections [77] and cancer [78,79]. A distinction has been made between N1 with the “anti-tumorigenic” phenotype and N2 or “pro-tumorigenic” neutrophils [80]. N1 neutrophils promote CD8^+^ T cell recruitment and activation by producing T-cell-attracting chemokines, such as CCL3, CXCL9, and CXCL10, and pro-inflammatory cytokines, such as IL-12, TNF-α, and VEGF [81]. N2 neutrophils do not produce high levels of pro-inflammatory agents, but produce arginase, which would serve to inactivate T-cell effector functions in the same way as proposed for M2 TAMs [82].

In normal inflammatory situations, the inflammatory response is limited in time, and immune cells can resolve the situation [83], but, in malignant tissues, proinflammatory signals continue and intensify the response to satisfy tumor requirements. Using a zebrafish model of melanoma driven by HRAS^G12V^, wound-induced inflammation has been found to increase tumor formation in a neutrophil-dependent manner. Mechanistically, neutrophils are rapidly recruited from a wound to pre-neoplastic cells, increasing tumor cell proliferation [84].

The relevance of chemokine in neutrophil recruitment in melanoma has also been shown in zebrafish models. For example, the proinflammatory chemokine interleukin 8, which mediates neutrophil recruitment in the zebrafish inflammatory response [85], has been shown to have proangiogenic activity mediated by CXCR2 and to enhance melanoma cell invasion [86].

## 4. Crosstalk between Inflammation and NAPDH Oxidase-Derived Oxidative Stress in Melanoma

Oxidative stress represents an imbalance between the normal production of free radicals, such as reactive oxygen species (ROS), and the ability of the organism to detoxify the intermediates and repair the resulting damage [87]. ROS are products of cellular metabolism, most of them produced by the NADPH oxidases family, xanthine oxidoreductase, or the mitochondrial respiratory chain [88,89].

In a homeostatic state, ROS are essential for cell survival and normal cell signaling, preventing damage to cells. Antioxidants help maintain hydrogen peroxide (H_2_O_2_) levels for cellular signaling by reducing intracellular H_2_O_2_ to H_2_O. Some of the antioxidants involved in this process are catalase, glutathione peroxidase (GPX), peroxiredoxins (PRX), and thioredoxin (TRX) [90].

Chronic inflammation stimulates the production of ROS and nitrogen species (RNS), which mediates the recruitment of immune cells. These molecules induce DNA damage and mutation of oncogenes or tumor suppressor genes that can initiate the transformation of a cell into a malignant cell [60]. Macrophages, together with mast cells, eosinophils, and recruited neutrophils, increase the concentration of ROS and RNS in the tumor microenvironment, attempting to participate in the natural defense against infection [91]. However, increased concentration of ROS and RNS may have mutagenic consequences through DNA lesions or altered gene expression in proliferating cells.

ROS may be involved in different stages of melanoma development. Melanoma cells respond to hypoxia by stabilizing the hypoxia-inducible factor-1 (HIF-1a/b), which activates several genes that regulate important biological processes, such as cell proliferation, angiogenesis, metabolism, apoptosis, and migration [92]. Furthermore, tumor cells increase the production of ATP due to their increased metabolism. These higher levels of ATP can affect ROS homeostasis and modulate the activation of cell signaling pathways, leading to enhanced cell growth [93]. In addition, it has been shown that ROS produced by NADPH oxidase activates the master inflammatory transcription factor NF-κB and enhances melanoma cell proliferation [94]. Therefore, the complexity of biochemical networks makes it difficult to distinguish between the effects of ROS produced by proliferating cells and proliferation stimulated by ROS. However, it is clear that ROS are critically involved in the survival and proliferation of melanoma cells [95].

In melanoma, some evidence suggests that oxidative stress plays an important role in inhibiting metastasis. Melanoma cells in the blood experienced oxidative stress that was not observed in subcutaneous tumors. A hostile environment may hinder the migration process of metastasizing melanoma cells. These cells undergo reversible metabolic changes during metastasis that allow them to adapt to survive in conditions of oxidative stress, including an increased dependence on NADPH generating enzymes in the folate pathway. Thus, oxidative stress limits distant metastasis of melanoma cells in vivo, raising the possibility that treatment with antioxidants may favor the progression of this cancer by promoting metastasis [96].

It has been shown that immune cells, such as neutrophils and macrophages, are attracted to transformed cells at surprisingly early stages. An important attractant molecule is H_2_O_2_ [97], which is also an essential early damage signal responsible for driving neutrophils to wounds [98,99,100]. H_2_O_2_, which is produced by both transformed cells and their healthy neighbors, can diffuse away from its site of generation and may act as a signaling factor (Figure 3). Furthermore, it is becoming clear that hydrogen peroxide plays a fundamental role in cell proliferation, migration, and metabolism, as well as cell death [101].

Although the NAPDH oxidase dual oxidase 1 (DUOX1) was first described as a source of H_2_O_2_ in the thyroid, new data suggest that DUOX1 is one of the main sources of H_2_O_2_ in several epithelial cell types [102]. Recently, the expression of DUOX1 has been reported in epidermal keratinocytes [103]. It has been demonstrated that the lack of DUOX1, and consequently its production of H_2_O_2_, affects the differentiation, adhesion, and junction mechanisms in normal human keratinocytes. Thus, DUOX1 alterations in the epidermis could contribute to skin pathologies such as atopic dermatitis or psoriasis [104].

When H_2_O_2_ synthesis is blocked, pharmacologically or by knockdown of Duox1, the number of neutrophils and macrophages attracted to transformed cell clones is reduced, resulting in a decrease in the number of transformed cells [97]. These results suggest that innate immune cells play a supporting role in early transformation. In addition, proinflammatory macrophages without DUOX1 expression show an improved antitumor response by increasing the production of some proinflammatory cytokines, such as TNFα, IFNγ, and CXCL9, among others [105].

Prostaglandin E2 (PGE_2_) has been shown to be the trophic signal required for this expansion of transformed cells at very early stages of cancer initiation. The cyclooxygenase-2 (COX-2)/PGE2 pathway is critical in the earliest stages of tumor development [106]. This prostaglandin can promote a macrophage M2 phenotype and can redirect dendritic cells toward a myeloid-derived suppressor cell phenotype [107,108]. Leukocytes produce PGE_2_ as trophic support for the growth of transformed cells. This reveals a key contribution of host immune cells to the optimal growth of transformed cells in these early stages of cancer initiation.

Furthermore, a reduction of PGE_2_ levels by inhibiting COX-2 leads to a change in macrophage behavior with increased proinflammatory activity, which results in a more active engagement and engulfment of transformed cells in a zebrafish model of early cell transformation using oncogenic HRAS^G12V^ [106]. It could be very interesting to study the relevance of this signaling pathway in melanoma using recent advanced tools such as a *crestin* reporter line or MiniCoopR plasmid [49,51].

Another important observation made in vivo for the first time using the transparency of zebrafish is that oncogenic RAS induces ROS that eventually leads to a DNA damage response and aberrant cell proliferation [109] (Figure 4). Mechanistically, RAS-induced ROS are produced by NAPDH oxidase 4 (NOX4) and ROS scavengers, and NOX4 inhibition rescues a cell from death and larval malformations promoted by the overexpression of oncogenic RAS [109]. These results may be of clinical relevance since NOX4 is increased in human pancreatic tumors, and specific inhibition of NOX4 with small molecule inhibitors act synergistically with chemotherapeutic agents in mouse models of this type of cancer [109].

## 5. Conclusions

The contribution of ROS generated by NAPDH oxidases to cancer initiation, progression, and metastasis, as well as its crosstalk with inflammation, is largely unknown. Recent data point to an opposing contribution of ROS to each of these stages in melanoma, highlighting the need for further research using a combination of animal models and clinical samples. The zebrafish provides a complementary model that can help shed light on these complex mechanisms, especially at the initiation phase, thanks to its easy genetic and pharmacological manipulation, together with its optical transparency and specific tools to identify early transformation, such as the *crestin* reporter for melanoma initiation.

## Figures and Tables

**Figure 1 antioxidants-11-01277-f001:**
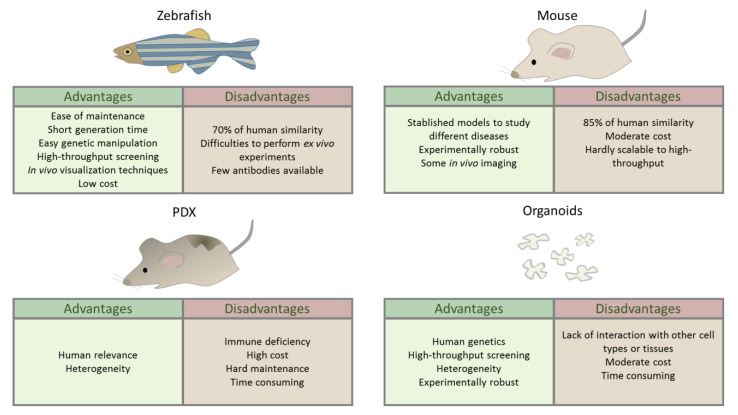
Advantages and disadvantages of zebrafish model compared with mouse, patient-derived xenograft (PDX), and human organoid models.

**Figure 2 antioxidants-11-01277-f002:**
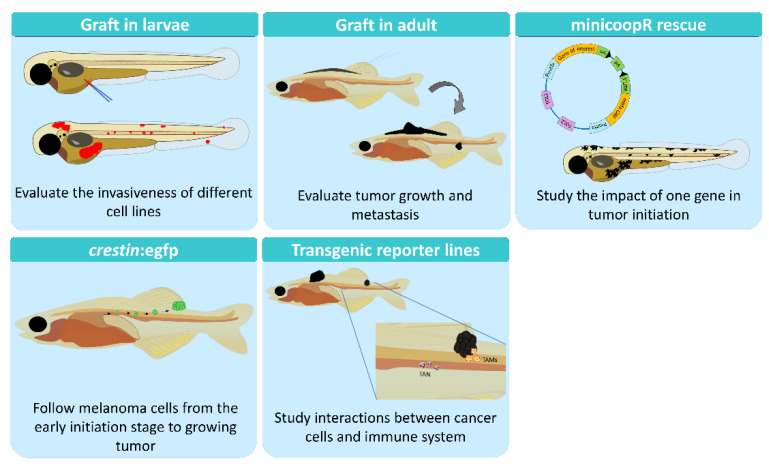
Zebrafish tools to study melanoma in vivo. GOI, gene of interest.

**Figure 3 antioxidants-11-01277-f003:**
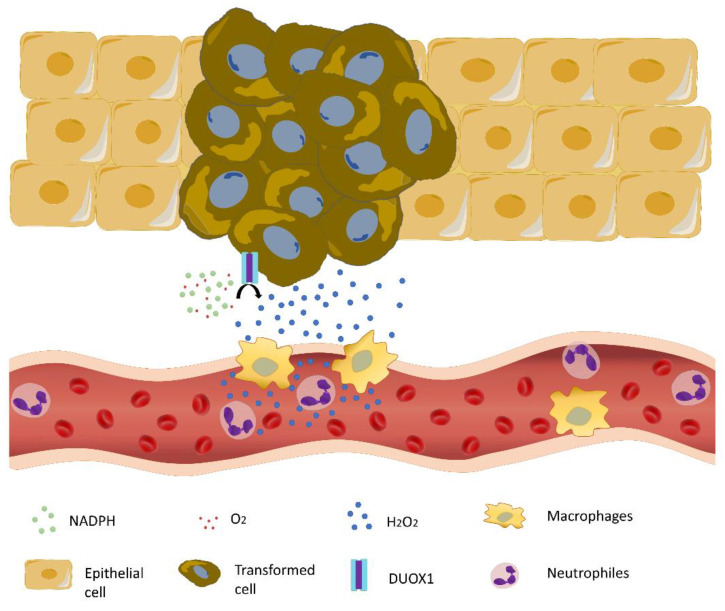
DUOX1-derived hydrogen peroxide from transformed cells and their neighboring cells attracts immune cells facilitating tumor growth.

**Figure 4 antioxidants-11-01277-f004:**
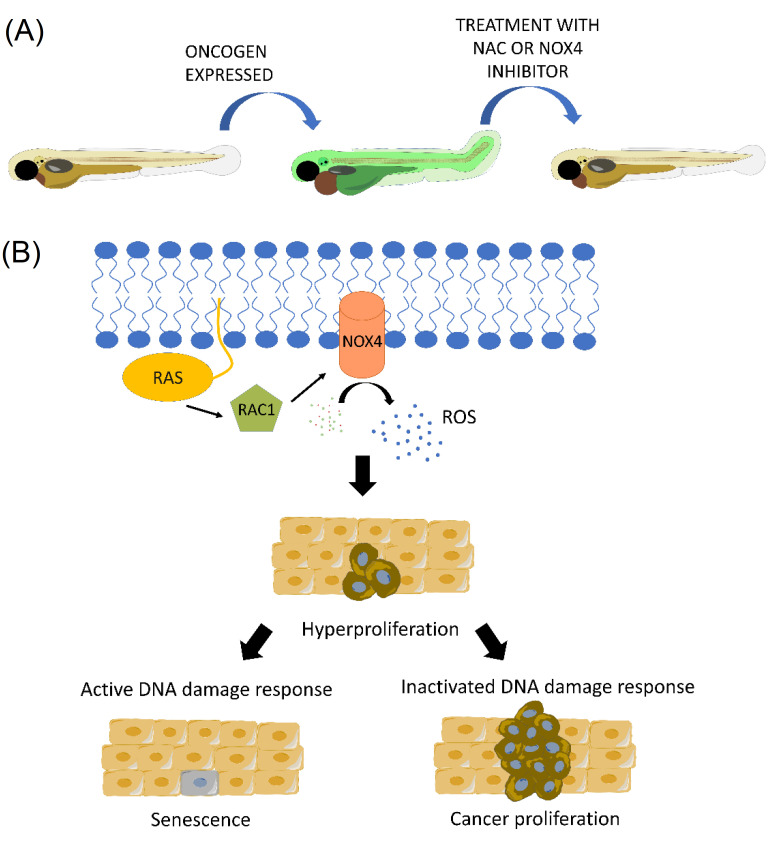
(**A**) ROS scavengers and NOX4 inhibition rescue cell from death and larval malformations originated by the overexpression of oncogenic RAS; (**B**) RAS-induced ROS are produced by NOX4 and induce tumor hyperproliferation by dependent and independent DNA damage response mechanisms.

## Data Availability

Data is contained within the article.

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
