# Peer review of "Zebrafish Models to Study the Crosstalk between Inflammation and NADPH Oxidase-Derived Oxidative Stress in Melanoma"

_antioxidants, 2022, doi:10.3390/antiox11071277_

Round 1

Reviewer 1 Report

Dear authors,

this work is an important contribution to the field, and it clearly demonstrates how important model organisms are for medical research. The only main minor criticism that I have has the role of ROS (especially NOX4 derived) in tumorgenesis in its focus. In line 299 and at many other instances it is mentioned that ROS could be mutagenic and thus promotes tumorgenesis. This is eventually true for the ETC (due to the close proximity to mtDNA) or exogenous (UV) induced ROS production. In case of NADPH oxidase derived ROS I somehow doubt this. Most NAPDH oxidases are far from the nucleus and DNA (NOX4 is complicated because it was found in the PM, mitochondria and ER). ROS have a very low half-time and therefore act only at a very short distance!!!! Therefore it is eventually better to clearly state that ROS (especially H2O2) are a second messengers and mitogens (facts that get more and more accepted). Accordingly also Figure 3B should be adapted. In this figure it looks like that NOX4 is somehow apart from the membrane, but in fact is an integral part of it.

Minor minor points:

Line 31: Exact values/perecnetages should also be  given for people above age 50.

Line 91: post fertilization are two words

Line 147: format issues

Line 208: proposed

Author Response

This work is an important contribution to the field, and it clearly demonstrates how important model organisms are for medical research. The only main minor criticism that I have has the role of ROS (especially NOX4 derived) in tumorgenesis in its focus. In line 299 and at many other instances it is mentioned that ROS could be mutagenic and thus promotes tumorgenesis. This is eventually true for the ETC (due to the close proximity to mtDNA) or exogenous (UV) induced ROS production. In case of NADPH oxidase derived ROS I somehow doubt this. Most NAPDH oxidases are far from the nucleus and DNA (NOX4 is complicated because it was found in the PM, mitochondria and ER). ROS have a very low half-time and therefore act only at a very short distance!!!! Therefore it is eventually better to clearly state that ROS (especially H2O2) are a second messengers and mitogens (facts that get more and more accepted). Accordingly also Figure 3B should be adapted. In this figure it looks like that NOX4 is somehow apart from the membrane, but in fact is an integral part of it.

Thanks a lot for your comments. We have revised the text and we think we clearly discuss the relevance of hydrogen peroxide as a second messengers and mitogens. However, we would like to also highlight the ability of this molecule to induce DNA damage. As regards, Fig. 3, it summarizes the results of reference 109, where it is reported that NOX4-derievd ROS mediate both hyperproliferation and DNA-damage response activation. We have modified it to clearly show that NOX4 is embedded in the cell membrane.

Minor minor points:

Line 31: Exact values/percentages should also be  given for people above age 50.

Line 91: post fertilization are two words

Corrected.

Line 147: format issues

Fixed.

Line 208: proposed

Fixed.

Reviewer 2 Report

Comments

This is a well described review on cross talk between inflammation and NADPH oxidase-derived oxidative stress in melanoma using zebrafish model. However, there are few points that would be proper to point out.

1. There are few grammatical errors in some sentences, kindly recheck it.

2. Check for the spelling errors in the figure.3.

3. It would be better if the specific site (organ) of injection of melanoma cells could also be mentioned in the section “zebrafish as a research model”.

4. Kindly add a section that describes which model organisms have already been used to investigate this crosstalk, and why zebrafish makes a unique model study this.

Author Response

This is a well described review on cross talk between inflammation and NADPH oxidase-derived oxidative stress in melanoma using zebrafish model. However, there are few points that would be proper to point out.

We are pleased with the reviewer’s comments.

  1. There are few grammatical errors in some sentences, kindly recheck it.

We have carefully read the manuscript and corrected all grammatical error identified.

  1. Check for the spelling errors in the figure.3.

Fixed.

  1. It would be better if the specific site (organ) of injection of melanoma cells could also be mentioned in the section “zebrafish as a research model”.

This is information has been added in sections 2.3 and 2.4.

  1. Kindly add a section that describes which model organisms have already been used to investigate this crosstalk, and why zebrafish makes a unique model study this.

Thanks for this important suggestion. We have added this comparison in section 2 and a new figure added (Fig. 1).

Round 2

Reviewer 2 Report

I recommend accepting this manuscript as it is revised carefully